# Determinants of the Adoption of Climate-Smart Agricultural Practices by Small-Scale Farming Households in King Cetshwayo District Municipality, South Africa

**Victor O. Abegunde [1], Melusi Sibanda [1,\*] and Ajuruchukwu Obi [2]**

[1] Department of Agriculture, University of Zululand, KwaDlangezwa 3886, South Africa;
overcomers001.av@gmail.com

[2] Department of Agricultural Economics and Extension, University of Fort Hare, Alice 5700, South Africa;
aobi@ufh.ac.za

\* Correspondence: SibandaM@unizulu.ac.za; Tel.: +27-(0)35-902-6068

**Abstract:** Agriculture, particularly small-scale farming, is both a contributor to greenhouse gas (GHG) emissions and a victim of the effects of climate change. Climate-smart agriculture (CSA) offers a unique opportunity to adapt to the effects of climate change while at the same time mitigating GHG emissions. The low response to the adoption of CSA among small-scale farmers raises questions as to the factors influencing its adoption in the small-scale farming system. With the aid of a close-ended questionnaire, structured interviews were conducted and formed the basis on which data were generated from 327 small-scale farmers selected through random sampling. Descriptive statistics, Composite Score Index and a Generalized Ordered Logit Regression (gologit) model were employed for the analysis. The majority (56.6%) of the sampled farmers fell in the medium category of users of CSA practices, while the lowest proportion (17.7%) of the sampled farmers fell in the high category. The use of organic manure, crop rotation and crop diversification were the most popular CSA practices among the sampled farmers. Educational status, farm income, farming experience, size of farmland, contact with agricultural extension, exposure to media, agricultural production activity, membership of an agricultural association or group and the perception of the impact of climate change were found to be statistically significant and positively correlated with the level of CSA adoption. Furthermore, off-farm income and distance of farm to homestead were statistically significant but negatively correlated with the CSA level of adoption. This paper argues that climate change-related education through improved extension contact and exposure to mass media can strengthen integrated farm activities that bolster farm income. Additionally, farmer associations or groups should be given adequate attention to facilitate CSA adoption as a means to climate change mitigation and resilience.

**Keywords:** adaptation; climate change; climate-smart agriculture; small-scale farming; mitigation

## 1. Introduction

The climate-related shocks in agricultural production that farmers have to deal with have pushed the need for resilience in agriculture to the forefront of agricultural policies across the world [1]. There is a growing interest among policymakers and development practitioners to get as many farmers, mainly small-scale farmers, as possible to embrace sustainable farming practices that will fortify agricultural and food systems.

Climate change, to a great extent, has been the outcome of the accumulation of greenhouse gas (GHG), which is caused by human activities [2]. Agricultural activities such as bush burning and

deforestation are part of the human activities contributing to GHG emission [3,4]. Climate change has globally influenced natural and social systems [3]. However, studies have shown that developing countries, particularly African countries, are more vulnerable to the impacts of climate change [5–7]. The small-scale farming system in African countries is predominantly rain-fed, thereby making it highly susceptible to climate change and variability [8,9].

Many approaches have been recommended for mitigating the impacts of climate change on agricultural production. Climate-smart agriculture (CSA) practices that integrate the benefits of a sustainable increase in agricultural productivity, the adaptation and building of resilient agricultural and food security systems, as well as the reduction of GHG emissions from agricultural activities, have appeared to be very promising [10,11]. Researchers are left with the responsibility of informing policymakers. Policymakers need to be advised on the factors that can influence the adoption of these practices so that they can enact well-informed and practicable strategies that will facilitate the successful adoption of these practices by farmers.

Several studies have been carried out to elucidate the factors influencing the adoption of CSA practices by farmers [9,12–14]. Ojoko et al. [12] revealed that education, membership of a social group and access to credit were significant determinants of CSA adoption in Sokoto State in Nigeria. Akrofi-Atiotianti et al. [9], while investigating CSA adoption among cocoa farmers in Ghana, found that age and location of farms, farmers' age, residential status and access to extension services influence CSA adoption in the cocoa farming system in Ghana. Aryal et al. [14] studied the factors influencing the adoption of CSA practices by farmers in the Indo-Gangetic plains of India. They noted that farmers' characteristics such as gender, education, social and economic capital, as well as farmers' experience of climate risks and access to extension services and training were critical determinants of CSA adoption among the farmers. Murray et al. [13], in their study, argued for more attention to gender analysis while addressing the development and adoption of CSA tools and technologies.

However, the majority of the studies on CSA adoption only focused on one strategy. The concept of CSA encompasses a set of practices that farmers adopt in different combinations [1,15]. Furthermore, given the location and content-specific attributes of CSA application as regards to the economic, environmental and social situations, there is the need for location-specific studies on CSA. However, there is little information concerning CSA adoption in South Africa, particularly as it relates to the small-scale farming system.

According to Vera et al. [1] and Teklewold et al. [16], farmers enjoy more benefits when they adopt multiple strategies, as some of the strategies can be complementary to one another and enable the farmers to exploit relevant synergies. As a result, the adoption of multiple CSA practices helps in building a sustainable agricultural system that is very resilient to shocks which are related to climate change and other factors posing challenges to agricultural production.

This paper aims to bridge the information gap on CSA adoption in small-scale agriculture by assessing the factors affecting the level of adoption of CSA practices by small-scale farmers, using data collected from small-scale farmers in King Cetshwayo District Municipality (KCDM) of KwaZulu-Natal (KZN) Province of South Africa. Following the studies of Vera et al. [1], Ojoko et al. [12], Teklewold et al. [16] and Wekesa et al. [17], the agricultural practices adopted by the farmers, which have been identified to fit into the profile of CSA [18,19], were considered in the study to investigate the level of adoption of CSA practices among the sampled farmers.

## 2. Materials and Methods

### 2.1. Selection and Description of the Study Area

King Cetshwayo District Municipality is found in the KZN Province in South Africa, which covers an area of 94,361 $km^2$ and shares 7.7 per cent of South Africa's total area [20]. The KZN Province has a record of high agricultural production and the highest number of farming households in South Africa [21]. King Cetshwayo District Municipality is situated in the north-eastern region of

the KZN Province, constituting five local municipalities namely; uMhlathuze, Nkandla, uMfolozi, Mthonjaneni and uMlalazi. Figure 1 is a map of KCDM in relation to Mthonjaneni and uMhlathuze Local Municipalities selected for comparative analysis.

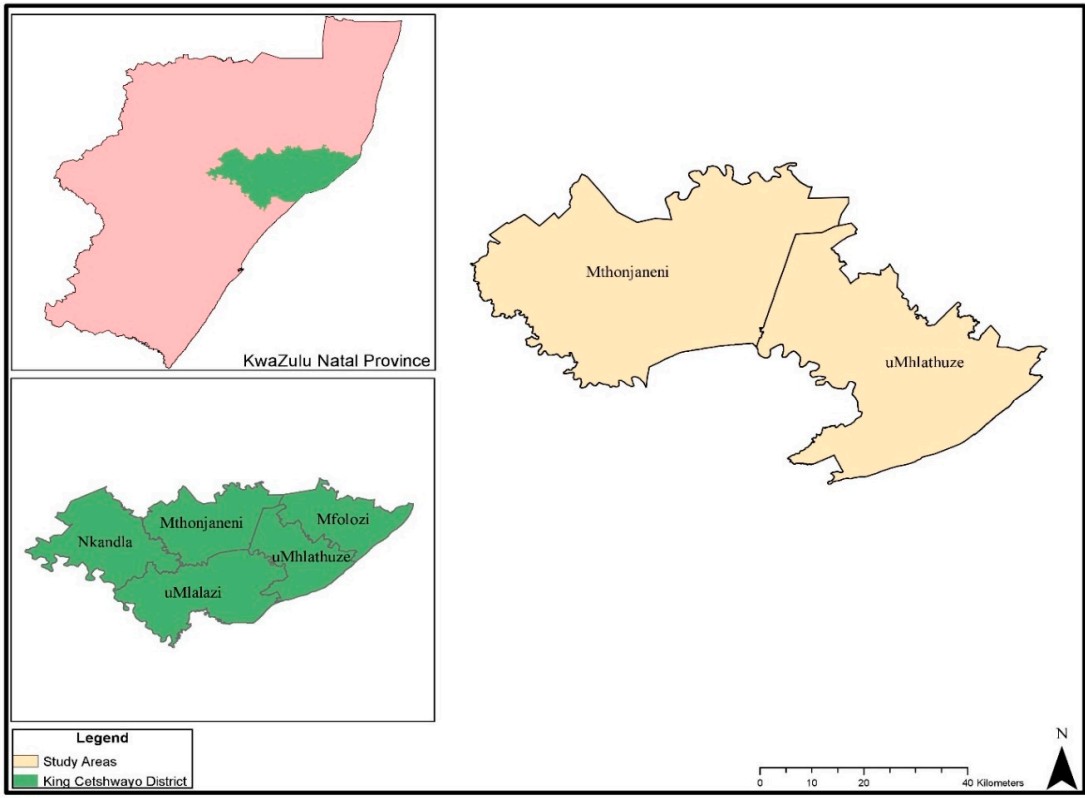

**Figure 1.** Map of King Cetshwayo District Municipality (KCDM).

A multi-stage sampling procedure was used for the selection of the location in which units of observation (study) have the required characteristics [22]. The multi-stage sampling technique involved the purposive selection of Mthonjaneni and uMhlathuze Local Municipalities based on their agricultural potential. The uMhlathuze Local Municipality has the agricultural sector with the lowest economic contribution (3.2%), while Mthonjaneni has the agricultural sector with the highest economic contribution (33.5%) [23].

*2.2. Research Design*

This paper adopted a cross-sectional research design with a quantitative research method. A cross-sectional research design is sufficient for the analysis of, and drawing of inferences from, existing differences between people, subjects or phenomena. Cross-sectional research designs can employ data from diverse backgrounds of disciplines and contrasting observational studies. It is comparatively less expensive to use and less time consuming [24,25]. The cross-sectional research design was considered to be appropriate for this study because it enables rational and sound conclusions, coupled with its time and cost-efficiency. The rationale for the choice of this research design is to obtain an accurate investigation into the research problem to gain a better insight into the factors influencing the adoption of CSA practices in the small-scale agricultural system, even with time and money constraints. Furthermore, the adoption of a cross-sectional design allows the use of a structured interview using close-ended questionnaires as a research instrument for this paper. The structured interview through the close-ended questionnaires enhanced the collection and conversion of data into a numerical form for statistical analysis and drawing of inferences. This design gives empirical evidence

on the underlying factors influencing the CSA adoption level. This paper utilizes both descriptive and inferential analytical techniques for robust conclusions.

### 2.2.1. Theoretical Framework

This study borrows from the theoretical framework of the theory of utility. As stated by Terdoo and Adekola [26], deciding whether or not to adopt any CSA practice falls under utility and profit maximization theoretical frameworks. The theory of utility explains the behavior of individuals on the basis that individuals can consistently rank their choices based on their preferences [27]. With the theory of utility, what is deemed necessary about utility concerning choice/s being made is whether an option has a higher utility than another and not the measure of the difference between the available options. The consideration of choices made on which agricultural practices to be adopted by farmers hangs on the concept of ordering available options based on the benefits they stand to receive from the practices.

There is the assumption that economic agents, including small-scale farmers, adopt CSA practices when the expected utility or net benefit is significantly higher than when they do not adopt [26]. As utility cannot be directly observed, the activities of economic agents could be observed through their choices. Consider a rational farmer whose aim is to maximize the proceeds from production over a specific period and has a set of CSA practice $j$ options to choose from. The farmer $i$ decides to adopt CSA practice $j$ if the utility from $j$ is perceived to be more than that from other options (assume, $k$). This relationship is expressed as Equation (1):

$$U_{ij} = (\beta'_j X_i + \varepsilon_j) > U_{ik} (\beta'_k X_i + \varepsilon_j), \ k \neq j \tag{1}$$

where $U_{ij}$ and $U_{ik}$ denote the perceived utility by farmer $i$ from CSA practice options $j$ and $k$, respectively; $X_i$ is a vector of regressors that influence the CSA practice option the farmer chooses; $\beta'_j$ and $\beta'_k$ are parameters of the independent variables; and $\varepsilon_j$ and $\varepsilon_k$ are the error terms, which based on an econometric assumption are independently and identically distributed [28–30].

Under the preference assumption that the farmer decides to adopt a CSA practice option which is more beneficial or generates net benefits and does not practice otherwise, the observable discrete choice of practice can be related to the latent continuous net benefit variable as Equation (2):

$$Y_{ij} = 1 \text{ if } U_{ij} > 0 \text{ and } Y_{ij} = 0 \text{ if } U_{ij} < 0. \tag{2}$$

In the generated formula, Y is a binary dependent variable valued as 1 when the farmer opts for a CSA practice and 0 if otherwise. The probability that farmer $i$ will choose CSA practice option $j$ among the set of adaptation options could be expressed as Equation (3):

$$(X = 1/X) = P (U_{ij} > U_{ik}/X) = P\left(\beta'_j X_i + \varepsilon_j - \beta'_k X_i - \varepsilon_k > \tfrac{0}{X}\right) = $$
$$P\left(\beta'_j X_i + \varepsilon_j - \beta'_k X_i - \varepsilon_k > \tfrac{0}{X}\right) = P(\beta* X_i + \varepsilon* > 0/X) = F(\beta* X_i) \tag{3}$$

where P is a probability function; $\varepsilon^* = \varepsilon_j - \varepsilon_k$ is a random disturbance term; $\beta^* = (\beta'_j - \beta'_k)$ is a vector of unknown parameters that can be explained as the net influence of the determinants of the choice of CSA practice; and $F(\beta^* X_i)$ is a cumulative distribution of $\varepsilon^*$ estimated at $\beta^* X_i$ [29–31].

### 2.2.2. Conceptual Framework

The main goals of CSA are a sustainable increase in agricultural productivity, climate change adaptation and reduction or total removal of GHG emissions [10,11]. The concept of CSA in its framework aims to integrate these three goals to enhance agricultural productivity and food security under a changing climate [10]. Alongside innovations and technologies that are being harnessed towards the implementation of CSA, agricultural practices that fit into the CSA framework or profile have been identified as CSA technologies [17]. Therefore, this study conceptualizes agricultural

practices that have been identified to fit into the CSA framework as CSA practices, and the level of adoption of these practices as the level of CSA adoption among the small-scale farmers in the study area/s. Farmers were asked to list the practices they adopt for their agricultural production. Agricultural practices which fit into the CSA profile based on FAO recommendation and the literature were identified for the study. Table 1 summarizes the CSA practices adopted by the sampled farmers.

**Table 1.** Definitions of CSA (climate-smart agriculture) practices understudy.

| CSA Practice | As Defined in This Study |
|---|---|
| Conservation agriculture | Reducing the extent of soil disturbance and allowing crop residue on the field after harvesting to protect the soil [32] |
| Agroforestry | Planting of trees or shrubs in or around farmland or pastureland [33] |
| The use of organic manure | Improving soil fertility through the manures and other organic sources [34] |
| Crop rotation | Planting different crops on the same area of farmland in consecutive planting seasons |
| Crop diversification | Planting varieties of crops of the same or different species [35] |
| Mulching | Covering the soil between plants with layer/s of material [36] |
| Use of wetland | Planting on wetlands where the soil is rich in water and fertility [37] |
| Use of drought and heat tolerant crops | Planting drought and heat tolerant crops [12] |
| Use of cover crops | Planting cover crops [12] |
| Soil conservation techniques | Preventing soil loss from erosion or alleviating soil fertility loss by reducing contamination through acidification, salinization or other chemicals [12] |
| Integrated crop-livestock management | Integrating crop and livestock production |
| Improved grazing | Efficient pasture management (improved sowing varieties of pasture, rotational grazing) [38] |
| Efficient manure management | Efficient management of manure on livestock production [38] |
| Diet improvement for animals | Improving the diet of animals to achieve more protein production with less feed and lower emission [38] |

The assessment of adoption intensity is often based on the relative area, but it is difficult to assess the exact area under each CSA practice. Following Aryal et al. [14] and Teklewold et al. [16], this paper assessed adoption level by the number of CSA practices adopted in the farmland of each sampled farming household. The farming households were categorized into low, medium, and high categories of users of CSA practices based on the range of CSA practices adopted.

The adoption of agricultural innovations or technologies among farmers, mainly small-scale farmers, is not automatic; there are vital players and actors involved in the small-scale farming system. Since the agricultural practices considered in this paper are those that fit into the CSA framework, the factors considered in terms of adoption are considered concerning CSA adoption. The characteristics of small-scale farmers, coupled with CSA technological options and agents of information at the disposal of the farmers, are factors to be considered in CSA implementation and adoption. Not only do these factors influence adoption, but they can also influence the level of adoption or adoption intensity. Farmers' characteristics can significantly influence adoption and adoption level, but the CSA technological options available could also be a significant determinant in CSA adoption level. Considering the available CSA practices or technological options of CSA becomes imperative in analyzing CSA adoption level, given the location-specific attribute of CSA. Figure 2 summarizes how this paper conceptualizes CSA adoption among small-scale farmers and how CSA adoption can be influenced.

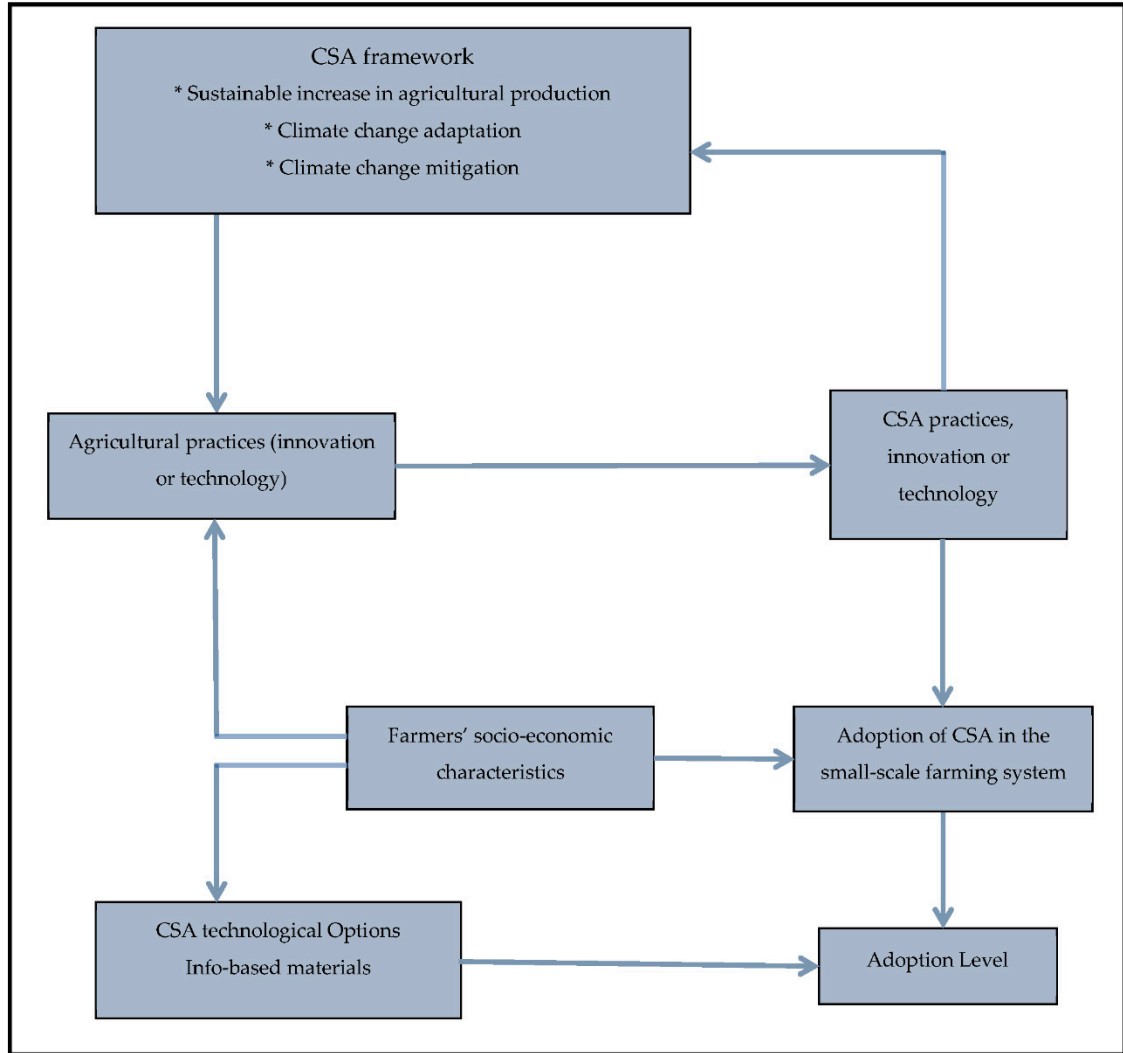

**Figure 2.** Conceptual framework—Adoption of CSA practices among small-scale farmers.

### 2.3. Description and Selection of Respondents

The target population in this paper were the small-scale farming households in KCDM. To ensure representative sampling, villages in the two chosen local municipalities were selected based on their agricultural potentials, based on the records from the Department of Agriculture and Rural Development (DARD) from each municipality. The records from the DARD in both local municipalities reveal that there are thirteen villages with approximately 720 small-scale farming households in Mthonjaneni Local Municipality and eight villages with approximately 1440 small-scale farming households in uMhlathuze Local Municipality. By implication, there is an estimated total of 2160 small-scale farming households in the two municipalities. A total sample size of 327 small-scale farming households used in this paper is derived from this population through a sample size calculator using a 95 per cent confidence level and confidence interval of 5. Following Hoyle et al. [39] in determining the sample size, further calculations were done to break down the sample size for each local municipality as follows:

Mthonjaneni Local Municipality: $n$ = (720/2160) * 327 = 109 small-scale farming households.
uMhlathuze Local Municipality: $n$ = (1440/2160) * 327 = 218 small-scale farming households.

Therefore, 109 and 218 small-scale farming households were derived for Mthonjaneni and uMhlathuze Municipalities, respectively. A random selection of farming households was then carried

out from the villages in these municipalities. A random selection gave every farming household in the two municipalities an equal chance of being selected.

*2.4. Data Collection*

A survey method was used for data collection through a structured interview using close-ended questionnaires. The questionnaires were used to elicit detailed information for descriptive and inferential analysis to reveal empirical evidence from the study. The questionnaires were pre-tested through a pilot study conducted with 35 small-scale farmers. This sample size is in line with Moore et al. [40] and Connelly [41] who recommend a 10 per cent of the actual sample size for a pilot study. The questionnaires were pre-tested for adequacy, appropriateness and effectiveness of the questions in capturing data needed for analysis. The time taken to complete the questionnaire was also given attention to while pre-testing the questionnaires. The respondents of the pilot survey did not participate in the actual study. The researcher also conducted a detailed focus group discussion to gain a better understanding of the typology (homogeneity/heterogeneity) of the target group and also identified vital issues affecting CSA adoption in the system.

Information gathered for farm activities and related data was for the previous farming season to ensure uniformity, regularity and completeness of the dataset. Questionnaires were conducted at the household level. The actual respondents were the person(s) responsible for household farming activities for each household. This approach helped to obtain a robust response and quality information on the agricultural activities of the farming households. Data were collected on the socio-economic characteristics of farmers (variables regarded as a set of farmers' features included age, gender, household status, level of education and income), CSA practices, and the yield from agricultural production among others.

The questionnaires were interviewer-administered to guide against the problem of misinterpretations or misunderstandings of words or questions. Data were collected during working hours (08h00–16h00), and unusual periods such as funerals, weddings, feasting events, and social grant pay-out days were avoided. Before the study, ethical clearance was obtained from the University of Zululand Research Ethics Committee (UZREC). The questionnaire was translated into the native isiZulu language in the study area/s for effective communication and to obtain relevant, credible and reliable information from respondents. The respondents were requested to sign informed consent forms before they took part in the research.

*2.5. Data Analysis*

The factors influencing the level of use of CSA by the small-scale farming households in the study area/s were analyzed using the composite score index and the gologit model. The composite score was computed from the responses of the farmers on the level of use of CSA practices. The farmers were then categorized into high, medium and low users with the use of a composite score. The category of the farmers from the composite score is used as a dependent variable in this paper, with hypothesized factors as independent variables in a logit model.

2.5.1. The Composite Score

This paper uses the composite score to determine the level of use of CSA practices by the sampled farming households in the study area/s. A binary scale that is using 1 to represent yes and 0 to represent no, concerning the use of the previously mentioned CSA practices, are used in rating the responses of the farmers. With the 14 statements/responses, a respondent can only have 14 and 0 as maximum and minimum points, respectively. The respondents are then categorized into high, medium and low users with the use of a composite score [12].

High user = those whose points fall between 14 and (Mean + S.D) points.

Medium user = those between upper and lower categories.

Low user = those whose points fall between (Mean − S.D) and 0.

### 2.5.2. The Generalized Ordered Logit Regression Model

The Generalized Ordered Logit Regression model in this paper is used to determine the factors influencing the level of use of CSA by the sampled small-scale farming households in the study area/s.

#### The Proportional Odds Assumption

One of the underpinning assumptions for the ordered logistic and ordered probit regression models is that the relationship between each pair of outcome groups is the same. Invariably, ordered logistic and ordered probit regression models operate on the assumption that the coefficients describing the relationship between, for example, the lowest and all higher categories of the response variable are the same as that describing the relationship existing between the next lowest and all higher categories. This assumption is called the proportional odds assumption or the parallel regression assumption. However, when the proportional odds assumption or the parallel regression assumption is violated, a generalized ordered logistic model becomes a more appropriate model [42,43].

#### Generalized Ordered Logit Regression Model

A major comparative advantage of the generalized ordered logit regression model is that it can estimate two exceptional cases of the generalized model: the proportional odds model and the partial proportional odds model. Generalized ordered logit regression can estimate models with lesser restrictions than the proportional odds or parallel lines models estimated by the ordinary ordered logit regression, whose assumptions are often violated, but more parsimonious and interpretable than those predicted by non-ordinal methods, such as multinomial logistic regression [43,44]. The generalized ordered logit regression model is expressed as Equation (4):

$$P(Y_i > j) = g(X\beta_j) = \frac{\exp(\alpha_j + X_i\beta_j)}{1 + \{\exp(\alpha_j + X_i\beta_j)\}}, \ j = 1, 2, \ldots, M - 1 \tag{4}$$

where $M$ is the number of categories of the ordinal regressand. From the equation stated above, the probabilities that $Y$ will take on each of the values $1, \ldots, M$ are equal to

$$P(Y_i = 1) = 1 - g(X_i\beta_j) \tag{5}$$

$$P(Y_i = j) = g(X_i\beta_j - 1) - g(X_i\beta_j) \ j = 2, \ldots, M - 1 \ P(Y_i = M) = g(X_i\beta_M - 1) \tag{6}$$

$$Y_i = \beta_0 + \beta_1 X_1 + \beta_2 X_2 + \beta_3 X_3 + \cdots \beta_n X_n + \mu_i. \tag{7}$$

The dependent variable $Y_i$ = level of usage of climate-smart agricultural practices (high user = 3; medium user = 2; low user = 1). $X_1 \ldots X_n$ represents the explanatory variables; $\beta_1 \ldots \beta_n$ represent the parameters of the explanatory variables; and $\beta_0$ represents the intercept, while $\mu_i$ represents the error term.

#### Description of the Explanatory Variables Used in the Gologit Model

Table 2 is a summary of the explanatory variables inputted in the gologit model for the analysis of the determinants of the level of CSA adoption among the sampled small-scale farmers. These include respondents' age, educational status, household size, farm income, off-farm income, farming experience, distance of farm to homestead, size of farmland, contact with extension agents, exposure to media, gender, marital status, agricultural production activity, membership of agricultural-related group, and perception of the effect of climate change.

**Table 2.** Explanatory variables in the gologit model used for the analysis of the determinants of the level of CSA adoption and their expected outcomes.

| Variable | Description and Measurement Type | Expected Outcome (+/−) |
|---|---|---|
| Age | Age of respondent in years (continuous) | +/− |
| Educational status | Number of years spent by respondents in acquiring formal education (continuous) | + |
| Household size | Number of members in the household (continuous) | + |
| Average monthly farm income (US $) | Total monthly income from farm enterprise measured in South African Rand (ZAR) and converted to United States dollar (US $) at the rate of ZAR 1 to $US 0.068 as of November 2019 for analysis purposes (continuous) | + |
| Average monthly off-farm income (US $) | Total monthly income from non-farm enterprise, measured in ZAR and converted to US $ (continuous) | +/− |
| Farming experience | Number of years spent in farming (continuous) | + |
| The distance of farm to homestead | The distance of home to the farm location in kilometers (continuous) | − |
| Size of farmland | The size of farmland in hectares (continuous) | + |
| Contact with extension agents | Number of yearly visits of extension agents (continuous) | + |
| Exposure to media | Number of mass media outfits accessed (continuous) | + |
| Gender | Respondents' gender (male = 1; female = 0) (dummy) | + |
| Production activity | Nature of farming activity (mixed farming = 1; one enterprise activity = 0) (dummy) | + |
| Membership to an agricultural-related group | Whether the respondent belongs to an agricultural-related group or association (yes = 1; otherwise = 0) (dummy) | + |
| Perception of the effect of climate change | Perception of respondent on the impact of climate change (significant adverse impact = 1; otherwise = 0) (dummy) | + |

Source: Authors (2019).

The explanatory variables inputted in the model specification were included based on the theoretical framework and empirical evidence from the literature [14,45]. Studies have shown that the lumpiness of and the complementarities among some explanatory variables constitute a package of actions which must be in place to get the desired outcome in CSA adoption [46]. The influence of the characteristics of farming households on adoption decisions is made pronounced by institutional failure and market asymmetry [47]. The direction of the influence of age on CSA adoption level could not be predetermined as older farmers are expected to have more farming experience with the likelihood of higher accumulation of physical and social capital.

On the other hand, ageing is associated with a decline in physical ability, increase in risk averseness and short term planning. Literacy can be associated with off-farm income and a higher level of exposure to new information and the capacity to process acquired information. This situation increases the likelihood of a farmer to adopt CSA technology or practices. Ngema [48] stated that an educated farmer is expected to approach farming with insights from exposure to information as a result of literacy. The household size in this paper represents the number of people residing together in a home at the time of this study. A large household increases a household's labor endowment. Although migration of household members may result in higher access to alternative sources of income, it reduces the number of potential labor in the household. Household members can constitute family labor, wherein important farming activities can be delegated to them [48]. It is hypothesized that more household members will facilitate CSA adoption, particularly in relation to labor-intensive practices.

Wealthier households are less risk-averse and have a higher capacity for risks related to the adoption of practices or technologies [45]. A farmer with a high-income level can afford to adopt

new or recommended practices based on the ability to finance the activities involved. Farm income was, therefore, predetermined to drive the CSA adoption level in a positive direction. The influence of off-farm income could not be predetermined. Access to alternative sources of income could translate into an increased capacity for the adoption of practices or technologies for households, since such households may enjoy better exposure to information and be able to finance investments. On the other hand, employment outside farming activities may reduce the time and effort devoted to farming activities, causing a reduction in labor endowment and investments in the adoption of new or recommended practices or technologies [12,49].

The more the farming experience, the higher the likelihood of accumulation of physical and social capital. The accumulation of physical and social capital can offer farmers better exposure and capacity to adopt new or recommended agricultural practices or techniques [47]. Farming experience is hypothesized to drive the CSA adoption level in a positive direction. Distance affects important factors such as access to technologies, information, credit facilities and implementation of techniques [16]. The distance of the farm to the homestead is expected to negatively correlate with the level of CSA adoption, as longer distances can reduce farmers' drive to try out new or recommended innovations or techniques. Additionally, farmland that is at a long distance from farmers' residence will cost more in terms of transportation of inputs and will be more difficult to monitor. Size of farmland has been found by some studies to influence the adoption of management practice positively [14,45]. It is hypothesized that farmers with larger farmland sizes are more likely to adopt more CSA practices. Access to agricultural extension and advisory services plays an important role in enhancing the adoption of agricultural innovations and techniques critical to agricultural production [50]. Onyeneke et al. [47] opined that extension services are essential sources of information for climate change adaptation and resilience. Contact with agricultural extension workers is anticipated to positively correlate with the level of CSA adoption. Adoption is influenced by access to information [51]. Sapkota et al. [52] noted that farmers with better exposure to farm-related information have a better disposition to the adoption of innovation or practices. Although farmers could obtain information from different sources, they typically make use of farm-related information which could be of benefit to their production. Exposure to media was predetermined to drive the CSA adoption level positively.

Kassie et al. [45] pointed out that women have less access to crucial farm resources such as land and labor and are faced with discrimination in terms of access to information and inputs, Simelane [53] noted that small-scale farming in South Africa is practiced primarily at the local level by women. However, Elias et al. [54] argued that equal participation of men and women in agriculture would enhance agricultural production. Given this arguments, the gender of the respondents was captured in the model as a dummy variable, where male equals a numeric value of 1, and 0 equals otherwise, with the effect of gender on the CSA adoption level predetermined to be positive.

Integrating crop and livestock production requires more monitoring and management practices, which could make farmers adopt more innovation or techniques. Production activity was captured as a dummy variable, where respondents practicing mixed farming are assigned a numeric value of 1, while those engaged in a one enterprise activity are assigned a value of 0. The effect of the nature of agricultural production activity of the respondents on the CSA adoption level was predetermined as positive. Membership to an agricultural-related group is taken as social capital, which could influence public spheres in rural areas. Belonging to a social group could facilitate access to needed information. Farmers who belong to the bottom of the social hierarchy have access to fewer sources of information and could have a lower perception of the adoption of innovations or techniques [51,55]. Membership to an agricultural-related group was anticipated to have a positive correlation with the level of CSA adoption. Farmers who experience high temperatures or low rainfall are more likely to engage in crop diversification and adopt CSA practices [14]. Besides, Vera et al. [1] noted that farmers who observed an increase in floods and changes in moisture levels in their area would likely adopt more CSA practices. Perception of the impact of climate change is anticipated to have a positive correlation with the CSA adoption level.

## 3. Results

### 3.1. Socio-Economic Characteristics of Respondents

Table 3 summarizes the socio-economic characteristics of the sampled small-scale farmers in the study area/s.

**Table 3.** Socio-economic characteristics of the sampled small-scale farmers in Mthonjaneni and uMhlathuze Municipalities.

| Variable | Mthonjaneni | | | | uMhlathuze | | | | Combined Analysis (KCDM) | | | |
|---|---|---|---|---|---|---|---|---|---|---|---|---|
| | Mean | Max | Min | SD | Mean | Max | Min | SD | Mean | Max | Min | SD |
| Age (years) | 48.2 | 72 | 29 | 13.1 | 51.7 | 78 | 30 | 10.7 | 50.5 | 78 | 29 | 11.1 |
| Education status (years) | 5.7 | 15 | 0 | 5.2 | 7.4 | 16 | 0 | 5.1 | 6.5 | 16 | 0 | 5 |
| Household size (number of members) | 8.7 | 18 | 5 | 3.5 | 8.7 | 17 | 3 | 3.1 | 8.7 | 18 | 3 | 3.2 |
| Farm income (US $) | 83.6 | 278.8 | 34 | 63.8 | 93.4 | 295.8 | 40.8 | 61.7 | 91.7 | 295.8 | 34 | 62.4 |
| Off-farm income (US $) | 164.4 | 340 | 34 | 208.4 | 177.5 | 353.6 | 54.4 | 138.1 | 178.3 | 353.6 | 34 | 164.7 |
| Farming experience (years) | 13.3 | 46 | 5 | 10.6 | 13.1 | 50 | 4 | 9.7 | 13.1 | 50 | 4 | 9.9 |
| Distance of farm to homestead (km) | 0.6 | 2.5 | 0.1 | 0.5 | 0.6 | 2.4 | 0.1 | 0.4 | 0.6 | 2.5 | 0.1 | 0.5 |
| Size of farmland (hectares) | 3.5 | 4.5 | 2 | 2.6 | 3.2 | 4 | 2 | 2.2 | 3.5 | 4.5 | 2 | 2.5 |
| Contact with extension agents (number of contacts) | 6.9 | 24 | 0 | 5.7 | 8.1 | 26 | 2 | 5.1 | 7.7 | 26 | 0 | 5.3 |
| Exposure to media (number of media outfits accessed) | 2 | 4 | 0 | 0.8 | 2.1 | 4 | 0 | 0.8 | 2.1 | 4 | 0 | 0.8 |
| Variable | Percentage | | | | Percentage | | | | Percentage | | | |
| Gender | | | | | | | | | | | | |
| Male | 44 | | | | 41.7 | | | | 42.5 | | | |
| Female | 56 | | | | 58.3 | | | | 57.5 | | | |
| Production Activity | | | | | | | | | | | | |
| Mixed farming | 58.7 | | | | 37.2 | | | | 38.5 | | | |
| One enterprise | 41.3 | | | | 62.9 | | | | 61.5 | | | |
| Membership of an agricultural-related group | | | | | | | | | | | | |
| Yes | 59.6 | | | | 45.9 | | | | 50.5 | | | |
| No | 40.4 | | | | 54.1 | | | | 49.5 | | | |
| Perception of the effect of climate change | | | | | | | | | | | | |
| Adverse effects | 56.9 | | | | 49.1 | | | | 51.7 | | | |
| No adverse impact | 43.1 | | | | 50.9 | | | | 48.3 | | | |

Source: Survey data (2018/19).

Results on the sampled farming households, as shown in Table 3, reveal that Mthonjaneni, uMhlathuze and KCDM (the combined analysis) were dominated by older farmers, with an average age of 48, 52 and 51 years respectively. The state of literacy of the sampled farmers reveal their average level of education to be at the primary level of education, since the average years of schooling of the farmers in Mthonjaneni was 6 years, while that of the farmers in uMhlathuze and KCDM was 7 years.

The sampled farming households from each area has an average of 9 members. The average size of the households shows the potential labor endowment in the study area/s. The members of the households are the potential labor force for the farming activities of the households, which could facilitate CSA adoption by the farming households, particularly CSA practices that are labor-intensive.

The average monthly farm income of the sampled farmers in Mthonjaneni (US $84), uMhlathuze (US $93) and KCDM (US $92) placed the farmers above the 2019 South African national poverty line (US $83). The average monthly income from sources other than farming (off-farm) activities reveals that the sampled households can generate more income from off-farm activities than farming activities.

The average years of farming experience in each of the study area/s were 13 years, which could be as a result of the dominance of older farmers in the study area/s. The average distance of farm to the homestead of the sampled farming households in Mthonjaneni, uMhlathuze and KCDM was 0.6 km. The average sizes of farmland of the sampled farmers were 3.5, 3.2 and 3.5 hectares in Mthonjaneni, uMhlathuze and KCDM, respectively. The average number of contacts with agricultural extension agents by the sampled farmers in Mthonjaneni in the farming season preceding the survey was 7 contacts, while the farmers in uMhlathuze and the combined analysis (KCDM) had 8 contacts. More contacts with agricultural extension and advisory services could enhance a more favourable disposition towards CSA adoption.

The sampled farmers' exposure to media was assessed by their exposure to radio, television, newspaper and internet. A numeric value of 1 is assigned to denote access to any of the media outfits considered. The sampled farmers in Mthonjaneni, uMhlathuze and the combined analysis (KCDM) were exposed to an average of two media outfits, out of the four presented to them.

Table 3 reveals the gender distribution of the sampled small-scale farmers in Mthonjaneni and uMhlathuze Municipalities. Female farmers accounted for the majority (56 and 58%) in Mthonjaneni and uMhlathuze Municipalities, respectively. The combined analysis also shows that females were the dominant (58%) group from the sampled farmers in KCDM. The majority (59%) of the farmers in Mthonjaneni Municipality practiced mixed farming, whereby they integrated livestock production with crop production, while the majority (63%) of their counterparts in uMhlathuze were involved in one enterprise activity, whereby they were solely into either crop or livestock production. The combined analysis reveals that the majority (62%) of the farmers were involved in one enterprise activity (Table 3).

Results on membership to an agricultural-related group suggest a stronger social network, concerning agricultural activities, among the sampled farmers in Mthonjaneni than their counterparts in uMhlathuze. There were more sampled farmers (60%) in Mthonjaneni who stated that they belong to one agricultural-related group or the other than those who stated that they do not belong to any agricultural-related group. In contrast, more than half of the sampled farmers (54%) in uMhlathuze stated that they do not belong to any agricultural-related group. The combined analysis (KCDM) reveals that a little above (51%) of the sampled farmers belong to one agricultural-related group or the other. Results in Table 3 suggest that there is more awareness of the adverse impact of climate change on agricultural production among the sampled farmers in Mthonjaneni than their counterparts in uMhlathuze. Slightly above half (52%) of the sampled farmers in KCDM are aware of the significant adverse impact of climate change on agricultural production.

### 3.2. Category of Respondents Based on the Level of Use of Climate-Smart Agricultural Practices

Results from the analysis of the frequency of use of the CSA practices among the sampled farmers reveal that the use of organic manure, crop rotation and crop diversification (in that order) were the most popular practices among the farmers in Mthonjaneni, uMhlathuze and the combined analysis (KCDM). In contrast, diet improvement for animals, agroforestry and the use of wetland (in that order) were the least popular CSA practices in Mthonjaneni Municipality, while efficient manure management, the use of wetland and agroforestry were the least popular in uMhlathuze. Results from the combined analysis show that the use of wetland, efficient manure management and diet improvement for animals (in that order) were the least popular practices among the sampled farmers in KCDM. Figure 3 illustrates the frequency of use of the CSA practices by the sampled farmers.

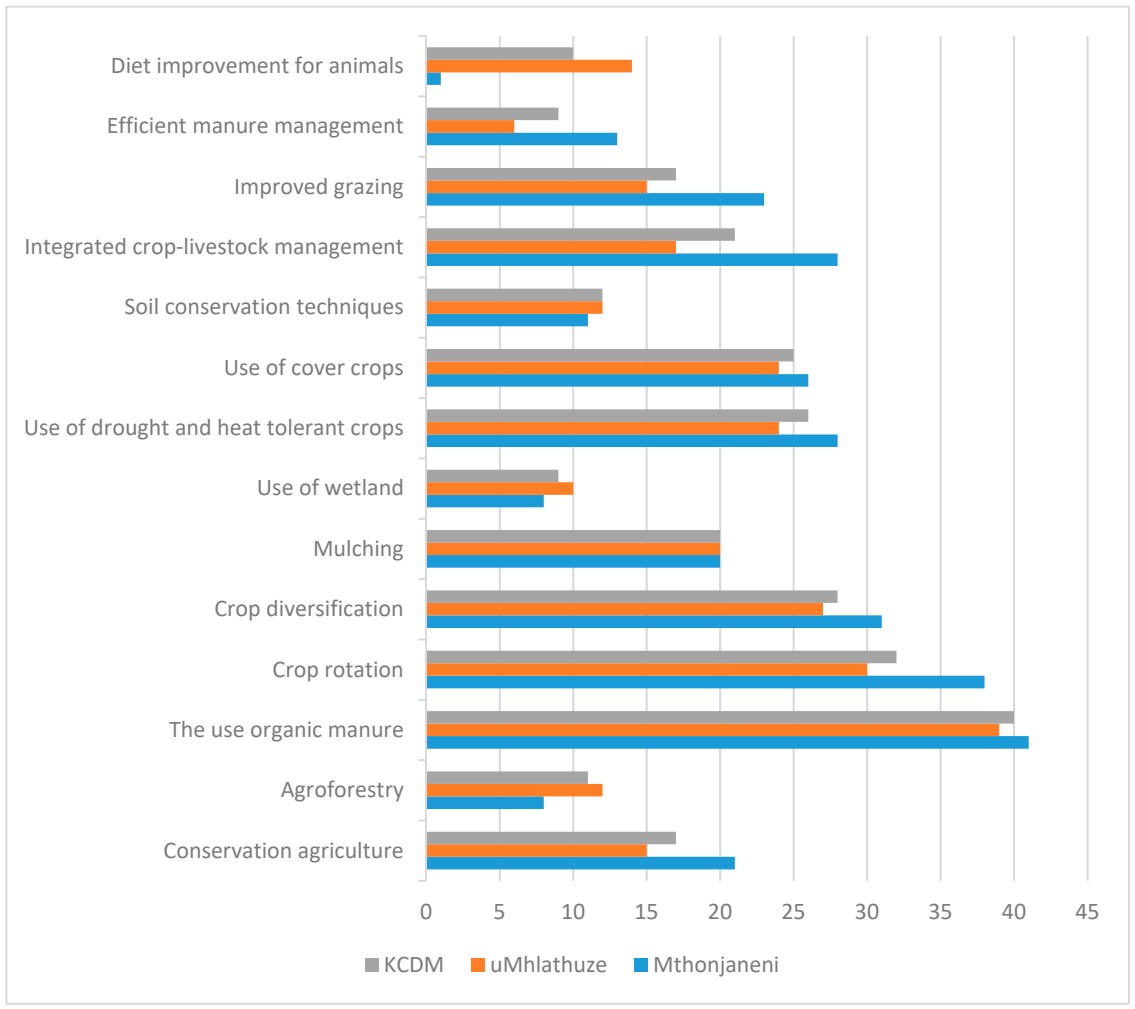

**Figure 3.** Frequency of use of CSA practices in Mthonjaneni, uMhlathuze and KCDM.

The composite score generated from the responses of the sampled small-scale farmers on their level of use of CSA practices was used to classify the farmers into three categories; that is low, medium and high users based on the frequency of use of CSA practices (Figure 3). Table 4 shows the category of users of CSA practices among the sampled farmers based on the number of CSA practices they adopted out of the identified CSA practices (see Table 1).

**Table 4.** Category of users of CSA practices among small-scale farmers in Mthonjaneni, uMhlathuze and KCDM.

| Category of Users of CSA Practices | Mthonjaneni | | uMhlathuze | | Combined Analysis (KCDM) | |
|---|---|---|---|---|---|---|
| | Frequency | Percentage | Frequency | Percentage | Frequency | Percentage |
| Low | 28 | 25.7 | 56 | 25.7 | 84 | 25.7 |
| Medium | 60 | 55.0 | 125 | 57.3 | 185 | 56.6 |
| High | 21 | 19.3 | 37 | 17.0 | 58 | 17.7 |
| Total | 109 | 100.0 | 218 | 100.0 | 327 | 100.0 |

Source: Survey data (2018/19).

Results from the composite score analysis reveal that more than half (55 and 57%) of the sampled small-scale farmers fell into the category of medium users in Mthonjaneni and uMhlathuze Municipalities, respectively. About 20 and 17 per cent were in the high user category in Mthonjaneni and uMhlathuze, respectively. About 26 per cent were in the lower user category in KCDM (both municipalities). Results from the combined analysis also reveal that the majority (57%) of the sampled small-scale farmers were medium users, while 26 and 18 per cent were low users of CSA practices in KCDM (Table 4).

*3.3. Factors Influencing the Level of Adoption of Climate-Smart Agricultural Practices*

3.3.1. Diagnostic Test

A diagnostic test was performed to check the proportional odds assumption in the models used. The violation of the proportional odds assumption makes a generalized ordered logistic model more appropriate than an ordered logit regression model or ordered probit regression model. A test was run for the model fit for Mthonjaneni, uMhlathuze and the combined analysis (KCDM). The null hypothesis is that there is no difference in the coefficients between models. Results on the approximate likelihood-ratio test of proportionality of odds across the response categories for Mthonjaneni (prob $> \chi^2 = 0.0193$), uMhlathuze (prob $> \chi^2 = 0.0358$) and in the combined analysis (prob $> \chi^2 = 0.0178$) indicate that there are significant differences in the coefficients in the models (Table 5). This observation implies that the proportional odds assumption or the parallel regression assumption has been violated. The results from the Brant test in Table 6 also establishes that the proportional odds assumption or the parallel regression assumption has been violated. Table 6 reveals the explanatory variables that could not still satisfy the parallel lines assumption after the variables have been constrained to have their effects satisfy the proportional odds assumption. Production activity ($p > \chi^2 = 0.040$) and perception of the impact of climate change ($p > \chi^2 = 0.004$) was found in the data analysis for Mthonjaneni Municipality. Given the violation of the proportional odds assumption, a generalized ordered logistic model is adopted.

**Table 5.** Likelihood-Ratio test of proportionality of odds.

| Variable | Mthonjaneni | | | uMhlathuze | | | Combined Analysis (KCDM) | | |
|---|---|---|---|---|---|---|---|---|---|
| | Coefficient | Std. Err. | $p > /z/$ | Coefficient | Std. Err. | $p > /z/$ | Coefficient | Std. Err. | $p > /z/$ |
| Age | −0.0043 | 0.0199 | 0.830 | 0.0065 | 0.0163 | 0.691 | 0.0114 | 0.0118 | 0.337 |
| Educational status | 0.0201 | 0.0533 | 0.706 | 0.0573 | 0.0315 | 0.170 | 0.0398 | 0.0250 | 0.112 |
| Household size | 0.0765 | 0.0790 | 0.333 | −0.0288 | 0.0558 | 0.605 | −0.0257 | 0.0426 | 0.546 |
| Average monthly farm income | 0.0065 | 0.0035 | 0.066 | 0.0008 | 0.0023 | 0.725 | 0.0027 | 0.0019 | 0.156 |
| Average monthly off-farm income | −0.0033 | 0.0016 | 0.037 | 0.0005 | 0.0010 | 0.996 | −0.0008 | 0.0007 | 0.260 |
| Farming experience | 0.0013 | 0.0227 | 0.953 | −0.0159 | 0.0163 | 0.329 | −0.0157 | 0.0125 | 0.211 |
| The distance of farm to homestead | −0.3306 | 0.4211 | 0.432 | −0.9032 | 0.3609 | 0.012 | −0.3840 | 0.2519 | 0.127 |
| Size of farmland | 0.0480 | 0.0996 | 0.630 | 0.0221 | 0.0881 | 0.802 | 0.0249 | 0.0652 | 0.703 |
| Contact with agricultural extension agents | 0.0398 | 0.0413 | 0.336 | 0.0243 | 0.0300 | 0.419 | −0.0077 | 0.0232 | 0.740 |
| Exposure to media | 0.0226 | 0.2779 | 0.935 | −0.0362 | 0.1739 | 0.835 | 0.0239 | 0.1432 | 0.867 |
| Gender | 0.2141 | 0.4165 | 0.607 | −0.2069 | 0.2776 | 0.456 | −0.1559 | 0.2212 | 0.481 |
| Production activity | 1.0163 | 0.2687 | 0.000 | 0.0286 | 0.1567 | 0.855 | 0.2534 | 0.1276 | 0.147 |
| Membership to an agricultural-related association or group | 0.1068 | 0.3623 | 0.768 | 0.1658 | 0.2783 | 0.551 | 0.0772 | 0.2095 | 0.712 |
| Perception of the impact of climate change | 1.0131 | 0.4461 | 0.023 | −0.0276 | 0.2858 | 0.923 | −0.2999 | 0.2300 | 0.192 |
| _ Cut1<br>_ Cut2 | 0.1663<br>3.7096 | 1.4562<br>1.5337 | (Ancillary<br>parameters) | −1.0705<br>1.7544 | 1.1370<br>1.1426 | (Ancillary<br>parameters) | −0.4611<br>2.3025 | 0.8517<br>0.8640 | (Ancillary<br>parameters) |
| Approximate likelihood-ratio test of proportionality of odds across response categories | $\chi^2(14) = 26.99$<br>Prob $> \chi^2 = 0.0193$ | | | $\chi^2(14) = 13.14$<br>Prob $> \chi^2 = 0.0358$ | | | $\chi^2(14) = 27.27$<br>Prob $> \chi^2 = 0.0178$ | | |

Source: Survey data (2018/19).

**Table 6.** Brant test of parallel regression assumption.

| Variable | Mthonjaneni | | | uMhlathuze | | | Combined Analysis (KCDM) | | |
|---|---|---|---|---|---|---|---|---|---|
| | $\chi^2$ | $p > \chi^2$ | df | $\chi^2$ | $p > \chi^2$ | df | $\chi^2$ | $p > \chi^2$ | df |
| | **25.15** | **0.033** | **14** | **12.42** | **0.572** | **14** | **29.37** | **0.109** | **14** |
| Age | 0.59 | 0.443 | 1 | 1.73 | 0.189 | 1 | 2.82 | 0.193 | 1 |
| Educational status | 0.73 | 0.392 | 1 | 0.91 | 0.341 | 1 | 1.77 | 0.184 | 1 |
| Household size | 0.18 | 0.669 | 1 | 0.73 | 0.394 | 1 | 0.39 | 0.533 | 1 |
| Average monthly farm income | 0.04 | 0.842 | 1 | 0.00 | 0.993 | 1 | 0.08 | 0.773 | 1 |
| Average monthly off-farm income | 2.38 | 0.123 | 1 | 0.46 | 0.496 | 1 | 4.12 | 0.142 | 1 |
| Farming experience | 0.15 | 0.697 | 1 | 0.86 | 0.353 | 1 | 0.60 | 0.440 | 1 |
| The distance of farm to homestead | 0.04 | 0.832 | 1 | 0.28 | 0.595 | 1 | 0.26 | 0.612 | 1 |
| Size of farmland | 0.39 | 0.533 | 1 | 1.06 | 0.304 | 1 | 2.00 | 0.612 | 1 |
| Contact with agricultural extension agents | 1.40 | 0.237 | 1 | 0.28 | 0.595 | 1 | 0.22 | 0.641 | 1 |
| Exposure to media | 2.68 | 0.102 | 1 | 1.62 | 0.203 | 1 | 4.16 | 0.141 | 1 |
| Gender | 0.68 | 0.409 | 1 | 0.00 | 0.985 | 1 | 0.16 | 0.687 | 1 |
| Production activity | 4.24 | 0.040 | 1 | 0.00 | 0.948 | 1 | 2.72 | 0.199 | 1 |
| Membership to an agricultural-related association or group | 0.33 | 0.565 | 1 | 0.00 | 0.969 | 1 | 0.39 | 0.531 | 1 |
| Perception of the impact of climate change | 8.16 | 0.004 | 1 | 3.41 | 0.065 | 1 | 9.89 | 0.102 | 1 |

A significant test statistic provides evidence that the parallel regression assumption has been violated. Source: Survey data (2018/19).

### 3.3.2. The Results from the Generalized Ordered Logit Regression

The dependent variable of the model is the category of users of CSA practices (high, medium and low), where the high user category is the reference category in the model. Internally, the generalized ordered logit regression generated constraints on the parameters. All the variables in the model fitted for the regression analysis of the responses from uMhlathuze and the combined analysis (KCDM) were constrained to have their effects satisfy the proportional odds or parallel lines assumptions. However, for the regression model fitted for the responses from Mthonjaneni, production activity and perception on the impact of climate change could not satisfy the parallel lines assumption. The difference is only in the interpretation of the parameters of those that do not eventually satisfy the parallel lines assumption. Interpretation is particularly straightforward for those variables that satisfy the parallel lines assumption.

Table 7 shows that educational status is statistically significant and has a positive influence on the level of adoption of CSA practices among the sampled small-scale farmers in the lower and medium user categories at 10 per cent ($p = 0.070$) level of significance in uMhlathuze Municipality. In contrast, it has no statistically significant influence in Mthonjaneni nor for the combined analysis (KCDM). Table 7 further shows that the average monthly farm income of the sampled farmers. This variable has a statistically significant and positive influence on the level of adoption of CSA practices in the lower and medium user categories in Mthonjaneni at 5 per cent level of significance ($p = 0.017$). However, average monthly farm income has no statistically significant influence in uMhlathuze nor for the combined analysis (KCDM). The average monthly off-farm income has a statistically significant and negative influence on the level of adoption of CSA practices at 5 per cent ($p = 0.038$) and 10 per cent ($p = 0.065$) level of significance of the sampled farmers in the lower and medium user categories in Mthonjaneni and uMhlathuze Municipalities, respectively (Table 7). Results from the combined analysis (KCDM) also show a statistically significant and negative influence at 5 per cent ($p = 0.027$) level of significance.

Results in Table 7 also reveal that farming experience has a statistically significant and positive influence on the level of adoption of CSA practices of the sampled small-scale farmers in the lower and medium user categories at 5 per cent ($p = 0.034$ and $p = 0.012$) level of significance in uMhlathuze and for the combined analysis (KCDM), respectively. However, farming experience has no statistically significant influence on the level of adoption of CSA practices in Mthonjaneni. The distance of the farm to homestead has a statistically significant and negative influence on the level of adoption of CSA practices of the sampled small-scale farmers in the lower and medium user categories in uMhlathuze at 1 per cent ($p = 0.012$) level of significance (Table 7). Results from the combined analysis (KCDM) show that the distance of the farm to homestead has a marginal statistically significance and negative influence at 10 per cent ($p = 0.104$) level of significance on the level of CSA adoption. The size of farmland statistically and positively influenced the level of adoption of CSA practices of the sampled farmers in the lower and medium user categories at 5 per cent level of significance in Mthonjaneni ($p = 0.028$), uMhlathuze ($p = 0.036$) and KCDM ($p = 0.045$).

**Table 7.** Results of the factors influencing the level of adoption of CSA practices by the respondents.

| Variable/Category | Mthonjaneni | | | uMhlathuze | | | Combined Analysis (KCDM) | | |
|---|---|---|---|---|---|---|---|---|---|
| | Coefficient | Std. Err. | $p > |z|$ | Coefficient | Std. Err. | $p > |z|$ | Coefficient | Std. Err. | $p > |z|$ |
| **Low** | | | | | | | | | |
| Age | −0.0057896 | 0.0209016 | 0.782 | 0.0064615 | 0.0162824 | 0.691 | −0.0036014 | 0.0135985 | 0.791 |
| Educational status | 0.0319474 | 0.0549755 | 0.561 | 0.0572535 * | 0.2776207 | 0.07 | 0.0086719 | 0.029313 | 0.767 |
| Household size | 0.0698497 | 0.083664 | 0.404 | 0.0288115 | 0.0557788 | 0.605 | 0.0248461 | 0.0426708 | 0.56 |
| Average monthly farm income | 0.0070143 ** | 0.0038846 | 0.017 | 0.0008007 | 0.0022758 | 0.725 | 0.0024858 | 0.0018836 | 0.187 |
| Average monthly off-farm income | −0.0034016 ** | 0.0016422 | 0.038 | 0.000051 * | 0.0010286 | 0.065 | −0.000742 ** | 0.0007567 | 0.027 |
| Farming experience | 0.0076485 | 0.025223 | 0.762 | 0.015906 ** | 0.0162956 | 0.034 | 0.0161027 *** | 0.0129847 | 0.012 |
| The distance of farm to homestead | −0.2614479 | 0.4573182 | 0.568 | −0.9031678 *** | 0.3609375 | 0.012 | −0.3989937 * | 0.2544398 | 0.104 |
| Size of farmland | 0.0450424 ** | 0.1007878 | 0.028 | 0.0221367 ** | 0.088111 | 0.036 | 0.0277942 ** | 0.064181 | 0.045 |
| Contact with agricultural extension agents | 0.0448575 ** | 0.0452909 | 0.021 | 0.0242936 ** | 0.0300432 | 0.042 | 0.0110755 ** | 0.0237835 | 0.053 |
| Exposure to media | 0.0448811 ** | 0.2840593 | 0.034 | 0.036175 ** | 0.1739412 | 0.049 | 0.039349 * | 0.1448349 | 0.063 |
| Gender | 0.2324539 | 0.4368736 | 0.595 | 0.2068779 | 0.2776207 | 0.456 | 0.1707497 | 0.2253137 | 0.449 |
| Production activity | 0.5520902 *** | 0.3177059 | 0.003 | 0.0285586 | 0.1566871 | 0.855 | 0.0925811 *** | 0.1524419 | 0.005 |
| Membership to an agricultural-related association or group | 0.09692 | 0.4208144 | 0.818 | 0.1658002 ** | 0.2782924 | 0.028 | 0.0601437 * | 0.216237 | 0.062 |
| Perception of the effect of climate change | 0.048473 ** | 0.6873162 | 0.038 | 0.0276045 * | 0.2858183 | 0.077 | 0.8065043 *** | 0.2802232 | 0.004 |
| Constant | 1.437723 | 1.635162 | 0.379 | 1.070446 | 1.136982 | 0.346 | 2.101383 | 0.9731928 | 0.031 |
| **Medium** | | | | | | | | | |
| Age | −0.0057896 | 0.0209016 | 0.782 | 0.0064615 | 0.0162824 | 0.691 | −0.0036014 | 0.0135985 | 0.791 |
| Educational status | 0.0319474 | 0.0549755 | 0.561 | 0.0572535 * | 0.2776207 | 0.07 | 0.0086719 | 0.029313 | 0.767 |
| Household size | 0.0698497 | 0.083664 | 0.404 | 0.0288115 | 0.0557788 | 0.605 | 0.0248461 | 0.0426708 | 0.56 |
| Average monthly farm income | 0.0070143 ** | 0.0038846 | 0.017 | 0.0008007 | 0.0022758 | 0.725 | 0.0024858 | 0.0018836 | 0.187 |
| Average monthly off-farm income | −0.0034016 ** | 0.0016422 | 0.038 | 0.000051 * | 0.0010286 | 0.065 | −0.000742 ** | 0.0007567 | 0.027 |
| Farming experience | 0.0076485 | 0.025223 | 0.762 | 0.015906 ** | 0.0162956 | 0.034 | 0.0161027 *** | 0.0129847 | 0.012 |
| The distance of farm to homestead | −0.2614479 | 0.4573182 | 0.568 | −0.9031678 *** | 0.3609375 | 0.012 | −0.3989937 * | 0.2544398 | 0.104 |
| Size of farmland | 0.0450424 ** | 0.1007878 | 0.028 | 0.0221367 ** | 0.088111 | 0.036 | 0.0277942 ** | 0.064181 | 0.045 |
| Contact with agricultural extension agents | 0.0448575 ** | 0.0452909 | 0.021 | 0.0242936 ** | 0.0300432 | 0.042 | 0.0110755 ** | 0.0237835 | 0.053 |
| Exposure to media | 0.0448811 ** | 0.2840593 | 0.034 | 0.036175 ** | 0.1739412 | 0.049 | 0.039349 * | 0.1448349 | 0.063 |
| Gender | 0.2324539 | 0.4368736 | 0.595 | 0.2068779 | 0.2776207 | 0.456 | 0.1707497 | 0.2253137 | 0.449 |
| Production activity | 0.825814 ** | 0.5620345 | 0.049 | 0.0285586 | 0.1566871 | 0.855 | 0.0925811 *** | 0.1524419 | 0.005 |
| Membership to an agricultural-related association or group | 0.09692 | 0.4208144 | 0.818 | 0.1658002 ** | 0.2782924 | 0.028 | 0.0601437 * | 0.216237 | 0.062 |
| Perception of the effect of climate change | 0.1810599 *** | 0.5953543 | 0.003 | 0.0276045 * | 0.2858183 | 0.077 | 0.8065043 *** | 0.2802232 | 0.004 |
| Constant | −6.273496 | 2.216619 | 0.005 | 1.070446 | 1.136982 | 0.346 | 2.101383 | 0.9731928 | 0.031 |
| Number of observations | 108 | | | 216 | | | 324 | | |
| LR $\chi^2$ (14) | 65.53 | | | 14.96 | | | 38.87 | | |
| Prob > $\chi^2$ | 0 | | | 0.0048 | | | 0.003 | | |
| PseudoR2 | 0.2953 | | | 0.2337 | | | 0.0616 | | |

\***, \*\*, \* denotes statistical significance at 1, 5 and 10% level. The high user category was generated as the reference category. Source: Survey data (2018/19).

Table 7 shows that contact with agricultural extension agents is statistically significant and positively influences the level of adoption of CSA practices among the sampled farmers in the lower and medium user categories at 5 per cent ($p = 0.021$, $p = 0.042$ and $p = 0.053$) level of statistical significance in Mthonjaneni, uMhlathuze Municipalities and for the combined analysis (KCDM), respectively. Table 7 further shows that exposure to media is statistically significant and positively influences the level of adoption of CSA practices among the sampled farmers in the lower and medium user categories at 5 per cent ($p = 0.034$ and $p = 0.049$) and 10 per cent (0.063) level of significance in Mthonjaneni, uMhlathuze Municipalities and for the combined analysis (KCDM), respectively. Results in Table 7 also show that production activity has a statistically significant and positive influence on the level of adoption of CSA of the sampled farmers in Mthonjaneni at 1 per cent ($p = 0.003$) and 5 per cent ($p = 0.049$) in the lower and medium user categories, respectively (the variable agricultural production activity for the sampled farmers in Mthonjaneni did not satisfy the parallel lines assumption). The agricultural production activity also has a 1 per cent ($p = 0.005$) level of statistical significance in the combined analysis (KCDM). In contrast, agricultural production activity has no statistically significant influence on the level of CSA adoption in uMhlathuze.

Membership of agricultural-related association or group was found to be statistically significant and positively influences the level of adoption of CSA practices among the sampled farmers in the lower and medium user categories in uMhlathuze and the combined analysis (KCDM) at 5 per cent ($p = 0.028$) and 10 per cent ($p = 0.062$) level of significance, respectively. However, this variable has no statistically significant influence on the level of CSA adoption in Mthonjaneni (Table 7). Results in Table 7 further shows that the perception of the effect of climate change has a statistically significant and positive influence on the level of CSA adoption in Mthonjaneni at 5 per cent ($p = 0.038$) and 1 per cent ($p = 0.003$) in the lower and medium user categories (the variable perception of the effect of climate change for the sampled farmers in Mthonjaneni did not satisfy the parallel lines assumption). Perception of the effect of climate change is also significant at 5 per cent ($p = 0.077$) and 1 per cent (0.004) level of significance in uMhlathuze and for the combined analysis (KCDM), respectively.

## 4. Discussion

The aim of this paper is to assess the factors affecting the level of adoption of CSA practices by small-scale farming households in KCDM, from two selected local municipalities, namely Mthonjaneni and uMhlathuze. Results on the category of the respondents based on their level of use of CSA practices reveal a similar pattern in both municipalities. The majority of the respondents fell into the medium user category, while the least proportion fell into the high user category. However, there is a higher proportion of respondents in the high user category in Mthonjaneni Municipality when compared to uMhlathuze Municipality. These findings suggest that the sampled small-scale farmers in Mthonjaneni Municipality are better adopters of CSA practices than those in uMhlathuze Municipality.

The results show a similar pattern in age distribution in the sampled small-scale farmers, revealing that there were more aged people involved in farming than youths in both municipalities. The low level of involvement of youths in agriculture could be because they find agriculture unattractive and prefer to search for jobs in other sectors. The dominance of older farmers in the farming system could be an advantage in terms of wealth experience and social capital in the system. However, the farming system could also be ladened with a state of languor.

Results on the educational status of the sampled farmers in uMhlathuze and Mthonjaneni reveal that they were exposed, on the average, to a primary level of education. The reports from the census conducted in 2001 and 2011, although low, reveal improved access to education in both municipalities over the years, but higher access to education in uMhlathuze than in Mthonjaneni [56]. Educational status was found to have a statistically significant and positive influence on the level of adoption of CSA practices in uMhlathuze Municipality, but did not influence CSA adoption level in Mthonjaneni and KCDM. Onyeneke et al. [47] and Onyeneke and Nwajiuba [57], in their studies, found that education positively affects climate change adaptation. Their findings agree with the findings of this paper on the

effect of educational status on CSA adoption level in uMhlathuze Municipality. A better level of literacy could be an added advantage in mainstreaming innovations and practices, particularly CSA among the farmers in the area. Farmers' literacy is expected to enhance their capacity to obtain, process and utilize information relevant to adoption and management of agricultural practices [47]. Additionally, farmers' education could enhance the productivity of agricultural extension services, as educated farmers could be more receptive to and productive with new agricultural innovations or practices. However, the result obtained is in contrast with the finding of Wekesa et al. [17], who reported a negative influence of years of schooling on the choice of CSA package. Wekesa et al. [17] argued that educated farmers would opt out of a CSA package if it does not offer risk reduction measures which could protect their investment against the risks of climate change.

The results on household size reveal a considerable average household size for the sampled small-scale farmers in both municipalities. A large household is a potential contributor to the labor force for agricultural activities, which the farming households in both municipalities could be enjoying [12]. Nonetheless, household size was found to have no statistical significance on the level of adoption of CSA practices in this paper.

The results from the combined analysis (KCDM) generally show that the sampled farmers earn fairly from their agricultural production when compared with the South African national poverty line, which is at an equivalent of US \$83.40 in 2019 [58]. However, results on average monthly farm income show that the sampled small-scale farmers in Mthonjaneni Municipality earned more than those in uMhlathuze Municipality. This finding corroborates the report on the agricultural potential of the two municipalities where Mthonjaneni is reported to have a higher agricultural potential [23]. Average monthly farm income has a statistically significant positive effect on the level of adoption of CSA practices in Mthonjaneni Municipality. Income from farming activities tends to increase the level of adoption of CSA practices by the farmers. This finding could be because farmers with higher farm income are less risk-averse and have better exposure to information [47,59]. This result is in line with the conclusion of Onyeneke et al. [47] who established that there is an increased likelihood for the adjustment of the agricultural production systems with an increase in farm income. Vera et al. [1], Wollni et al. [49] and Katengeza et al. [60] also confirm a statistically significant positive influence of farm income on the intensity of adoption of technologies and climate change adaptation. With an increase in income from farming activities, farmers would be able to acquire resources needed for the adoption of recommended practices or newly obtained information either from extension services, colleagues, social platforms or the media. Farmers need to be financially capable to adopt some agricultural practices or innovations successfully. Hence, financial empowerment is crucial to mainstream CSA adoption into the small-scale farming system successfully.

Results on income from other sources than farming (off-farm income) show that the sampled farmers from uMhlathuze Municipality had a higher average income from alternative sources than in Mthonjaneni Municipality. This finding could be because there are more economic activities in uMhlathuze Municipality, which could serve as sources of income more than in Mthonjaneni [23]. Such activities identified among the respondents include trading, handwork, cleaning and other sources of income such as child support and the old age grant. Average monthly off-farm income was found to have a statistically significant and negative influence on the level of adoption of CSA practices in this paper for both Mthonjaneni and uMhlathuze Municipalities. This finding is also true for the combined analysis (KCDM). The finding on off-farm income suggests that rather than using off-farm activities as mere alternative sources of income to augment income from farming activities, the off-farm activities were major sources of income for the farming households. A flourishing alternative source of income could result in a weak commitment to agricultural production. This result is in line with the findings of Vera et al. [1], who confirmed that income generation from other sources than farming reduces the likelihood of the adoption of more CSA practices. Farmers who can diversify their sources of income, generally can deal with agricultural production shocks and may see no need to improve the resilience of their agricultural production through the adoption of more CSA practices [1].

Generally, the results show a high farming experience for the sampled farmers in KCDM, which is in line with the descriptive evidence of the dominance of older farmers in the study area/s. However, there is a higher proportion of the sampled farmers with more years of farming experience in Mthonjaneni Municipality as compared to uMhlathuze Municipality. This could be one of the reasons for the higher agricultural potential reported for Mthonjaneni Municipality [23]. The farming experience was found to have a statistically significant and positive influence on the level of adoption of CSA practices in this paper for uMhlathuze Municipality. This finding is also true for the combined analysis (KCDM). This result implies that an increase in the years of farming experience would increase the level of CSA adoption. Farmers with more years of farming experience could have a higher level of CSA adoption through more efficient rapport with extension services and a stronger social network. According to Onyeneke et al. [47], farming experience significantly increases the likelihood of adjusting agricultural production and management systems. The stated result suggests that involving experienced farmers in promoting CSA among small-scale farmers can substantially impact the uptake of various CSA practices and enhance the implementation of CSA-related programs and projects among small-scale farmers.

The results on the distance of farm to homestead reveal the same average distance for Mthonjaneni, uMhlathuze and the combined analysis (KCDM). This finding could be attributed to the similar nature of settlement in both municipalities, which is expected since they are inhabited by the same group of people [21,23]. The distance of farm to the homestead of the sampled small-scale farmers in uMhlathuze Municipality has a statistically significant and negative effect on the level of CSA adoption. This finding is also true for the combined analysis (KCDM). This implies that farmers whose farms are far from their homesteads would likely adopt fewer CSA practices. This could be attributed to the challenges they would be facing in exercising proper and effective management as a result of stress posed by distance when compared with their counterparts who live closer to their farms. According to Teklewold et al. [16], besides influencing market accessibility, distance can also affect important factors critical to agricultural production such as access to technologies, information and credit institutions.

Size of farmland has a statistically and positive effect on the level of CSA adoption in Mthonjaneni, uMhlathuze and the combined analysis (KCDM). This finding indicates that farmers with larger farmlands adopted more CSA practices, thereby implying that land fragmentation could be a constraint to CSA adoption. Land is a crucial resource in agricultural production and farmers will be able to accommodate innovations or practices necessary for a successful agricultural venture with access to land and other needed resources.

Results on contact with extension agents reveal that the sampled farmers in uMhlathuze, on the average, had more extension contacts than their counterparts in Mthonjaneni Municipality. Access to extension services and advisory roles plays an essential role in the enhancement of adoption and innovation [61]. With increased and quality exposure to extension services in the farming system, farmers will likely have a higher chance of better exposure to new or more productive innovation or practices, particularly as it relates to CSA. Contact with agricultural extension agents has a statistically significant and positive effect on the level of adoption of CSA practices of the sampled small-scale farmers in both Mthonjaneni and uMhlathuze Municipalities. This finding is also true for the combined analysis (KCDM). Agricultural extension services serve as a crucial source of information on climate change, climate change adaptation and resilience, as well as agricultural management practices [47]. This result agrees with the finding of Onyeneke et al. [47], who confirmed that contact with extension agents increases the likelihood of the adoption of CSA practices. It is expected that frequent contact with extension agents would brighten the chances of awareness of climate change and CSA practices that can be adopted to adapt to climate variability and shocks. Besides, farmers, through extension services, can learn climate change mitigation measures and strategies that can enhance resilience. However, agricultural extension service, in influencing CSA adoption, can also be influenced by factors such as farmers' literacy, years of experience and financial capacity. Literacy and farming experience could enhance farmers' appreciation and processing of what extension service offers, while financial

capacity could influence the implementation. The complementarity of these factors forms a package that should be considered in mainstreaming CSA in the small-scale farming system.

Results on exposure to media show that the sampled farmers in Mthonjaneni, uMhlathuze and the combined analysis (KCDM), on the average, had the same level of exposure to media outfits. Exposure to media is statistically significant and positively affected the level of CSA adoption of the sampled small-scale farmers in both Mthonjaneni and uMhlathuze Municipalities. This finding is also true for the combined analysis (KCDM). Onyeneke et al. [47], in their study, also established that exposure to mass media increases the probability of the uptake of CSA practices. Birthal et al. [51], in their study, found that farmers making use of relevant information in agricultural production realized a 12 per cent higher net return per hectare compared with their counterparts. Farmers could learn about different innovations or practices through modern information, and make use of this information [52]. However, farmers' literacy could play a significant role in influencing farmers' exposure to information through the media. Exposure to media denotes that farmers who have considerable access to information through media have a better chance of awareness of the impacts of climate change and how to swiftly respond.

Results on gender show that a higher proportion of the sampled farmers in both municipalities were female. These results are consistent with the findings of Simelane [53], Kutya [62] and Masuku [63] who also observed that small-scale farming in South Africa is practiced primarily at a local level by older females. Nieuwoudt and Groenewald [64] also described small-scale farmers as usually consisting of elderly women. Both age and gender, however, were found to have no statistical significance on the level of adoption of CSA practices in this paper.

Generally, the results show that there was a higher proportion of the sampled farmers that were integrating crop and livestock production in Mthonjaneni Municipality as compared to uMhlathuze Municipality. Production activity was found to have a statistically significant and positive influence on the level of adoption of CSA practices in this paper for Mthonjaneni Municipality. This finding is also true for the combined analysis (KCDM). Results on the influence of production activity reveal that the sampled farmers who practice mixed farming would adopt more CSA practices than their counterparts who are engaged with only one enterprise activity. This finding suggests that small-scale farmers, who are more open to the integration of different farming techniques, would adopt more CSA practices compared to their counterparts who rely solely on one farming enterprise. Being open to the combination of varying farming techniques creates a promising and conducive platform for mainstreaming CSA [17].

Results on membership to an agricultural-related group show that there were more sampled farmers who belong to an agricultural-related group in Mthonjaneni than in uMhlathuze. The combined analysis indicates that there were more sampled farmers in KCDM who belong to an agricultural-related group than those who do not. Membership to an agricultural-related association or group was found to have a statistically significant and positive influence on the level of adoption of CSA practices in this paper for uMhlathuze Municipality. The result implies that the sampled farmers in both the lower and medium user categories who belong to an agricultural-related association or group would adopt more CSA practices than their counterparts who do not. Membership to a group is part of the build-up of the social capital of farmers, since it influences access to public spheres, particularly in rural areas [65]. Membership of agricultural groups plays crucial roles in the enlightenment of their members [12]. Agricultural associations or groups present a platform for farmers to discuss their challenges with their colleagues, thereby benefitting from counsel on how to cope with problems. Farmers with membership to an agricultural-related group could enjoy better access to information and resources [12]. A strong social network among farmers could enhance CSA adoption in the farming system.

Results on the perception of the sampled farmers of the effect of climate change show that there were more sampled farmers who perceived climate change to have an adverse effect on agricultural production in Mthonjaneni than in uMhlathuze. The combined analysis indicates that there were more farmers in KCDM who perceived climate change to have an adverse effect than those who did not.

Perception of the effect of climate change was found to have a statistically significant and positive influence on the level of adoption of CSA practices in this paper for Mthonjaneni and uMhlathuze Municipalities. This finding is also true for the combined analysis (KCDM). Results show that the sampled farmers who perceive that climate change has a significant adverse effect on agricultural production and food systems would adopt more CSA practices. This correlates with the finding of Vera et al. [1], who reported that farmers who observed an increase in floods and changes in moisture levels in their area had a higher probability of adopting more CSA practices. This finding is expected, considering that farmers who appreciate the risk attached to extreme weather conditions and varying weather patterns see the essence of adopting CSA practices to enhance their resilience to climatic shocks. However, based on the results, there is a difference in the matter of degree of the adoption practices. In Mthonjaneni Municipality, adopters are less likely to be in the low user category. The finding on the perception of the farmers of the effect of climate change suggests that an adequate level of awareness on the adverse impact of climate change on agricultural production among the farmers will likely enhance CSA adoption in the farming system of the area/s.

## 5. Conclusions

Few studies have been conducted to analyze the factors influencing the level of CSA adoption among South African small-scale farmers. This study fills this gap by identifying the factors affecting the level of adoption of climate-smart agricultural practices by small-scale farming households in the two selected municipalities in KCDM. The adoption of CSA practices, despite the benefits, is not automatic among small-scale farmers, hence the need to investigate the factors influencing CSA adoption in the small-scale farming system. This study found that factors such as education, farm income, income from non-farm sources, distance from home to farmstead, contact with agricultural extension agents, exposure to media, marital status, production activity, membership of an agricultural-related group and perception of the effect of climate change are significant determinants of CSA adoption level among small-scale farmers. The results of this study are of critical significance to the government, organizations and other stakeholders in South Africa with interest in the agricultural sector and in mitigating GHG emissions. The findings from this study have implications for further research and policy design and implementation.

The findings of this study can be used to inform policymakers and the DARD on best practices in mainstreaming CSA into the small-scale agricultural sector. Improving and strengthening contact with agricultural extension agents, increased exposure to media, and raising awareness on the impacts of climate change are critical in promoting the level of adoption of CSA practices. The implication of this is that adequate attention should be paid to extension services among small-scale farmers in South Africa, as frequent extension visits and services may primarily be very effective in mainstreaming the adoption of CSA practices among small-scale farmers. Furthermore, the use of media should be intensified in pushing awareness and information on the impact of climate change and CSA practices among small-scale farmers to help them cope and adjust.

Relevant stakeholders should endeavour to provide small-scale farmers with CSA-related extension messages. Furthermore, farmers can spread messages and information to their colleagues to ensure a full spread of information on climate change and CSA so that more small-scale farmers can adopt many CSA practices and techniques that will make their agricultural production systems more productive and resilient to climate change. Available platforms among small-scale farmers, particularly societies or groups, should be used to educate farmers about climate change and practices that can help them cope with challenges they face in agricultural production, concerning climate change.

A closer look at the results reveals that average off-farm income (income from sources other than farming) discourages the adoption of more CSA practices. This finding suggests that a stable income from alternative sources to farming may likely reduce the commitment of the small-scale farmers to farming activities, thus CSA adoption. Policies should be driven towards making agriculture

more attractive, while incentives should be given to encourage active participation and investment in agriculture, particularly to encourage small-scale farmers to adopt more CSA practices.

Concerning the challenge with distance to the farm from homesteads, farmers should be assisted with innovations and incentives that can enhance the proper monitoring of their farms. Furthermore, farmers should be supported with surmounting the challenge of transportation where it is an issue, to encourage easy access to their farms and markets for efficient productivity. The findings of the study on the size of farmland concerning the CSA adoption level suggests that land fragmentation could be a constraint to CSA adoption. Access to land would be an important consideration in mainstreaming CSA in the small-scale farming system. A great deal of attention should be given to integrating crop and livestock enterprises. This is critical in promoting CSA adoption. The implication of this is that small-scale farmers who practice mixed (integrated) farming are likely to be more open to various CSA practices than their counterparts who are involved in one farming enterprise activity.

The complementarities of some of the identified significant factors influencing the CSA adoption level in the farming system necessitates a package of actions, which are of policy implications. Efforts to improve the literacy level, exposure to media and financial capacity of farming households, and agricultural extension and advisory services are a package of actions, which should be considered in mainstreaming CSA in the small-scale farming system. While this paper determined the factors influencing the level of adoption of CSA practices among small-scale farmers in two selected municipalities in KCDM, it is recommended that researchers undertake further similar research on the factors influencing the level of CSA practices in other areas, but of in-depth enquiry, incorporating other indicators not considered in this paper, and with a comprehensive farm enterprise focus on each CSA practice.

**Author Contributions:** V.O.A. formulated the research investigation under the supervision of M.S. and A.O. V.O.A. undertook raw data collection, data analysis and draft manuscript compilation. Research scrutiny and scientific validation were carried out by M.S. and A.O. The outcome of the manuscript is the effort of all the authors. All authors have read and agreed to the published version of the manuscript.

**Funding:** This research was made possible with the financial assistance received from the "National Research Foundation of South Africa and The World Academy of Science (NRF—TWAS)" for NRF—TWAS African Renaissance Doctoral Scholarship (Grant UID 105460) and the University of Zululand Research and Innovation Office.

**Acknowledgments:** The authors would like to acknowledge NRF, TWAS and the University of Zululand Research and Innovation Office for providing financial assistance for this research project. The authors also extend gratitude to municipal authorities of the Mthonjaneni and uMhlathuze Local Municipalities and KCDM. Special thanks also go to the DARD, as well as the small-scale farmers in Mthonjaneni and uMhlathuze Municipalities for their participation in the study.

**Conflicts of Interest:** The authors declare no conflict of interest.

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
