# Peer review of "Determinants of the Adoption of Climate-Smart Agricultural Practices by Small-Scale Farming Households in King Cetshwayo District Municipality, South Africa"

_sustainability, doi:10.3390/su12010195_

Round 1

Reviewer 1 Report

There are two main issues to be highlighted.

First, the dependent variable is the number of CSA practices adopted by farmers. I would spend some words on justifying this methodological choice. In other words, I am not persuaded that the hypothesis of the higher the number of practices adopted the higher the degree of farmer's effort in the climate change mitigation is automatically demonstrated. You have to demonstrate that a small scale farmer, who probably does not have hectares enough for product diversification and then could not adopt all 14 practices taken into account, is not involved, or shows a lower degree of involvement, in CSA practices. On the other hand, one can adopt 11 CSA practices but his impact on climate change could be dramatically higher than the one produced by a specialized farmer who adopts only a few practices.

Second, the results are discussed under a reporting style without putting together the pieces of the puzzles. To be more precise, there is a strong hypothesis on the basis of the model, i.e. the independent variables are orthogonal to each other. In my opinion, you have to comment on the results putting together some of the relationships found. For example, extension service impact probably depends on the farmer's education degree and/or the farmer's age. Please make comments organic and hypothesize some "package" of variables that explain together the phenomenon. You can do it by using either some multivariate statistics methodology or, more simply, with an effort of identifying some relationships between the independent variables and spending some words on them.

Author Response

.

Reviewer 2 Report

Dear Authors,

Interesting research and presentation of work – well done. Below are comments intended to help improve development the paper. Please note that the comments below relate to very specific items with the piece:

P. 4, line 138: Please clarify what is meant by ‘parameters of the regression’. p. 4, lines 155-159: More supporting references are needed. P. 5, Figure 2: The figure does not make visual/conceptual sense. E.g., the three boxes/items left in the figure appear independent to the other data but then they, at points, influence this data. Rethink how this figure might be better presented. P. 7, lines 265-296: Does all this material need to be presented in the methods section? If possible include as an appendix item. Pp. 13-15, Table 4 and 5: A lot of material is presented in these tables. To help the non-statistical mined reader understand and interpret this data in: Table 4: highlight where the coefficients are different (e.g. bold text, highlighted in some way). Table 5: highlight where the “…parallel regression assumption has been violated”. Section 3.3.2: I am confused with the ‘p’ values discussed in this section of the piece; which causes me to doubt the data presented. Suggest making explicitly clear what ‘p’ value the results are refereeing to and exactly what section. Confirm the reporting of the ‘p’ values in the text as I could not find some data reported. E.g. p. 20, line 521: p = 0.035 (under Exposure to Media) could not be found in table 6. In table 6 the significance level is shown as ‘P’ but in the reporting of the findings it is shown as ‘p’ (i.e. P vs. p). Please be consistent, suggest changing to ‘p’ in table 6. With each reported result there is commentary at the end of each paragraph stating a possible implication. The implication(s) of the data is best reported in the paper’s discussion section and should be removed from this section. Section 4, p. 20: The reader needs to see the ‘higher use’ data or provide a detailed explanation of it in table 6. The existing sentence at the end of table (line 514) does not suffice. This is requested because this data is referred to and discussed extensively in the paper. P. 23, line 668-672: I question the discussion revolving around marriage as a factor. Please provide citations/refs. To help support this speculation or remove.

Once again good work and look forward to seeing the final version of the piece.

Author Response

.

Reviewer 3 Report

I have three principal observations to make about this paper:

1.  Perhaps the  most important point is the need for further discussion of the CSA practices involved.  There is a brief introduction (lines 83-87) but this would benefit from considerable expansion giving much greater detail on each practice.

Following on from this point is the need for the paper to explore which CSA practices were adopted and why.  The absence of this analysis is a major deficiency in the paper.

2.  Repetition.  The paper unnecessarily repeats much information which could be more concisely presented:

Lines 104-110 simply repeat lines 95-100. The independent variables listed in lines 276-291 are simply repeated in Table 1.  The same information is repeated again in lines 308-358.  I strongly suggest that this material is condensed and presented concisely, once, in an expanded Table 1. Lines 366-433 are a needless repetition of information presented in Table 2.  Again, the suggestion would be to present the information once, in an improved Table 2.

3.  Some of the descriptions of statistical methods could benefit from more detailed explanations.  For example:

Line 112, explain 'cross-sectional research'. Line 122 & 199, explain 'close-ended questionnaires. Line 451, explain 'proportional odds assumption'.

In general the descriptions of statistical methods are a little impenetrable (see, e.g., Sections 2.2.1 & 2.5.2) and should be made more clear and  concise if possible.

There are some additional observations and suggestions for improvement:

A thorough revision of the English throughout.  It's not bad but some improvements can be made. Figure 2 needs more explanation.  Lines 174-175 do not give enough. Table 2, add units in the first column. Table 3, give details of the CSAs used. Tables 4&6.  Use the symbol χ2. Table 1 (line 461) is Table 5.

Author Response

.

Round 2

Reviewer 3 Report

It seems that the authors have taken full note of the  the reviewers' suggestions for improvements.  In this case the paper should be considered for publication.

There are still some problems with the English (eg line 67 (practices), line 410 (under study), Table 1 (through), line 859 (these), Figure 3 units on horizontal axis, etc, etc, etc). but I assume the Sustainabilty editors will make the necessary corrections.